Motor self-regulation in goats (Capra aegagrus hircus) in a detour-reaching task

Langbein Jan langbein@fbn-dummerstorf.de
Institute of Behavioural Physiology, Leibniz Institute for Farm Animal Biology , Dummerstorf , Germany
Vallortigara Giorgio
Electronic publication date: 2018 Jul 5
Publication date: 2018
Volume: 6
Electronic Location ID: e5139
Received 2018 Mar 13; Accepted 2018 Jun 9
Copyright: ©2018 Langbein
Copyright year: 2018
Copyright holder: Langbein
License: This is an open access article distributed under the terms of the Creative Commons Attribution License, which permits unrestricted use, distribution, reproduction and adaptation in any medium and for any purpose provided that it is properly attributed. For attribution, the original author(s), title, publication source (PeerJ) and either DOI or URL of the article must be cited.
License URL: https://creativecommons.org/licenses/by/4.0/

Keywords: Goat, Motor self-regulation, Cylinder task, Inhibition, Detour task

Funding: The authors received no funding for this work.

==============================
Motor self-regulation is the ability to inhibit a prepotent response to a salient cue in favour of a more appropriate response. Motor self-regulation is an important component of the processes that interact to generate effective inhibitory control of behaviour, and is theorized to be a prerequisite of complex cognitive abilities in humans and other animals. In a large comparative study using the cylinder task, motor self-regulation was studied in 36 different species, mostly birds and primates. To broaden the range of species to comprehensively evaluate this phenomenon, motor self-regulation was studied in the domestic goat, which is a social ungulate species and moderate food specialist. Using the cylinder task, goats were first trained to perform a detour-reaching response to retrieve a reward from an opaque cylinder. Subsequently, an otherwise identical transparent cylinder was substituted for the opaque cylinder over 10 test trials. The goats’ ability to resist approaching the visible reward directly by touching the cylinder and to retain the trained detour-reaching response was measured. The results indicated that goats showed motor self-regulation at a level comparable to or better than that of many of the bird and mammal species tested to date. However, the individual reaction patterns revealed large intra- and inter-individual variability regarding motor self-regulation. An improvement across trials was observed only in latency to make contact with the reward; no improvement in the proportion of accurate trials was observed. A short, distinct pointing gesture by the experimenter during baiting did not have any impact on the side of the cylinder to which the goats detoured. In half of goats, individual side biases were observed when detouring to the side of the cylinder, but there was no bias at the population level for either the left or right side. The results underline the need for a detailed examination of individual performance and additional measures to achieve a complete understanding of animal performance in motor self-regulation tasks.

Introduction

In recent years, a large body of research has studied a behavioural phenomenon described as inhibitory control or self-control in various species of birds and mammals. This phenomenon is defined as an individual’s ability to inhibit an impulsive or prepotent response, normally in reaction to a salient cue or stimulus, in favour of a more appropriate response (Bray, MacLean & Hare, 2014; MacLean et al., 2014; Jelbert, Taylor & Gray, 2016; Vernouillet et al., 2016). This type of behavioural control has been discussed as one of the prerequisites for problem solving and is assumed to be an important aspect of complex cognitive capabilities, such as reasoning and planning (Kralik, Hauser & Zimlicki, 2002; Diamond, 2013), and regarded as essential for effectively interacting with the environment (Burke et al., 1991). Studies of different primate species have attributed well-developed inhibitory skills to living in complex social groups and have found that higher levels of fission–fusion dynamics are correlated with better inhibitory control and consequently higher behavioural flexibility (Amici, Aureli & Call, 2008; Maclean et al., 2013).

Inhibitory control is a core component of the so-called executive functions (Miyake et al., 2000). The executive functions (Diamond, 2013) comprise a cluster of top-down mental processes activated when a behaviour switches from automatic, instinctual or learned execution to high-level control (insight, inference or reasoning) associated with greater cognitive effort. According to Diamond (2013), inhibitory control is based on several subdomains: cognitive and attentional inhibition, which are subsumed under the category interference control, as well as response inhibition. Diamond (2013) equates the latter with behavioural inhibition or self-control. By contrast, Beran (2015) hierarchically separated response inhibition from self-control. According to his definition, response inhibition requires only the inhibition of a prepotent motor response, whereas self-control requires decision making as well. Other authors have classified response inhibition and self-control as separate processes under the generic term ’behavioural inhibition’ (Bari & Robbins, 2013).

The most frequently used paradigms to compare inhibitory control across different mammal and bird species are the A-not-B task (Osthaus et al., 2013; Nawroth, Borell & Langbein, 2015a), reversal-learning tasks (Tapp et al., 2003; Bond, Kamil & Balda, 2007), delay-of-gratification tasks (Anderson, Kuroshima & Fujita, 2010; Hillemann et al., 2014) and detour-reaching tasks (Kabadayi, Bobrowicz & Osvath, 2018). For the latter, either a transparent barrier (Pongracz et al., 2001; Vlamings, Hare & Call, 2010; Baragli et al., 2011) or a transparent cylinder (Bray, MacLean & Hare, 2014; MacLean et al., 2014) is often used which the animal must detour to reach a reward. In light of the debate regarding subdomains of inhibitory control (Diamond, 2013) as well as the debate about the classification of response inhibition and self-control, it is questionable whether the different tasks mentioned above all measure the same aspects of inhibitory control (Manrique & Call, 2015; Brucks et al., 2017). The A-not-B task is strongly dependent on selective attention as it is affected by such contextual features as the number of boxes presented, the speed of movements and visual distinctiveness (Kabadayi et al., 2016). Reversal-learning tasks and especially delay-of-gratification tasks have been widely accepted as accurately measuring self-control (Evans et al., 2014; Beran, 2015); however, performance may also be impacted by task-specific demands, which depend on other executive functions, such as working memory and cognitive flexibility (Manrique & Call, 2015).

This basic aspect of behavioural inhibition has recently been referred to as motor self-regulation (Bray, MacLean & Hare, 2014; Kabadayi et al., 2016), and this term is used hereafter in this paper. It has been argued that detour-reaching tasks, such as the cylinder task, likely reflect fundamental inhibitory skills, which require only the inhibition of a prepotent response elicited by a salient cue in favour of a more appropriate motor pattern (Bari & Robbins, 2013; Kabadayi, Bobrowicz & Osvath, 2018). In the cylinder task, subjects are first trained to locate a reward hidden inside a horizontally oriented, opaque cylinder that is open at both sides. To retrieve the reward, subjects must detour to the side of the cylinder. Once the animals are well trained in this task, an otherwise identical transparent cylinder is substituted for the opaque cylinder such that the subject can now see the reward when approaching. Subjects that show high levels of motor self-regulation are expected to detour to one of the open ends of the cylinder as previously learned without first touching the front of the apparatus. MacLean et al. (2014) have used the cylinder task to investigate motor self-regulation (not self-control as they claimed; see above Bari & Robbins, 2013; Beran, 2015) using a broad comparative approach. The focus of the MacLean et al. study, as with other studies applying the cylinder task, was on primate, canine and bird species (Marshall-Pescini, Viranyi & Range, 2015; Fagnani et al., 2016; Kabadayi et al., 2016; Vernouillet et al., 2016). The investigators found levels of motor self-regulation above 90% in the great apes and some social corvids, whereas performance was below 50% in many other bird and mammal species, including primates.

To broaden the range of species for a comprehensive discussion of behavioural inhibition, motor self-regulation was evaluated using the cylinder task in an ungulate species, the domestic goat. Feral and domestic goats have been shown to live in fission–fusion societies (Stanley & Dunbar, 2013; Ævarsdóttir, 2014) and show various features related to complex cognition, such as learning to learn (Langbein, Siebert & Nuernberg, 2008), categorization (Meyer et al., 2012), inferential reasoning (Nawroth, Borell & Langbein, 2014), object permanence (Nawroth, Borell & Langbein, 2015a) and learning of complex two-step tasks (Briefer et al., 2014). In this study, goat capacity for motor self-regulation was evaluated as well as whether goats show motor self-regulation spontaneously or learn to inhibit a prepotent response. While the experimental setup was identical in most respects to the one used by MacLean et al. (2014), it was also investigated whether short, distinct pointing gestures made by the experimenter during baiting would influence the side of the cylinder to which the goats detoured. Although three studies have shown that goats understand human gestures as an indication of a hidden reward (Kaminski et al., 2005; Nawroth, Borell & Langbein, 2015b; Nawroth, Borell & Langbein, 2016a), their understanding of distal human gestures has yet to be demonstrated. As there is some evidence that the lateralization of the brain underlies side biases in detour-reaching tasks (Vallortigara & Bisazza, 2002; Reddon & Hurd, 2009), it was investigated whether the goats would show a side bias in detouring the cylinder. Given their high level of sociability and their strong performance in a variety of cognitive tasks, goats can be expected to show a high level of motor self-regulation. According to the ‘grazer-browser continuum’ proposed by Hofmann (1989), the goat is classified as an ‘intermediate grazer’ that feeds on a mixture of shrubs/herbs/forbs and grass (Stuth, 1991), which might favour a high level of motor self-regulation to inhibit feeding on low-quality food in favour of searching for high-quality food.

Animals, Materials & Methods

Ethics statement

All animal care and experimental procedures were performed in accordance with the German welfare requirements for farm animals and the ASAB/ABS Guidelines for the Use of Animals in Research (Anonymous, 2016). All procedures involving animal handling and treatment were approved by the Committee for Animal Use and Care of the Ministry of Agriculture, Environment and Consumer Protection of the federal state of Mecklenburg-Vorpommern, Germany (Ref. No. 7221.3 − 2 − 012∕15 and 7221.3 − 2 − 011∕16 ).

Subjects and management conditions

The experiment was conducted with 22 female Nigerian dwarf goats (Capra aegagrus hircus) between April and June in 2015 (n = 10) and 2016 (n = 12). The goats were bred and housed at the Leibniz Institute for Farm Animal Biology (FBN, Dummerstorf, Germany). At the beginning of the experiment, all of the goats were between 15 and 22 months (mean, 17 months). The animals were group-housed indoors. Their pen (3 × 4 m) contained straw bedding and was equipped with an automatic waterer. The goats had ad libitum access to hay and were not food-restricted during any phase of the experiment. They were maintained under a photoperiod of 12 h light: 12 h dark, with the lights turned on at 6 am. During testing, the experimental area was supplemented with artificial light. All of the goats had participated in a study on visual discrimination learning using a fully automated learning device at the age of six months. The goats also underwent open-field, novel-object and maze tests at that time (S Osterwind & A Finkemeier, 2014/2015, unpublished data) and were thus already habituated to human handling before the start of the detour-reaching experiment.

General aspects of training and testing

The area for testing the goats was located in the same building as the holding pen and comprised several compartments: a waiting area, a start box, the experimental area, and two return alleys (Fig. 1). The walls of the experimental area and the return alleys were 1.6 m in height and made of brown plywood. The doors to and from the experimental area were operated remotely. The floor was covered with black rubber mats.

Figure 1 Sketch of the area for testing the goats.

The sketch shows the different compartments of the test area: a waiting area, a start box, an experimental area and two return alleys. The experimental area was video monitored. All doors to and from the experimental area were operated remotely.

For habituation, the goats were moved as a group to the waiting area and were allowed to enter the experimental area by passing through the open start box once a day for 4 consecutive days. They freely explored the experimental area and the return alleys as a group for 30 min each day. Training and testing were conducted in sessions from 9:00 to 11:00 and 13:00 to 15:00 from Monday to Friday. For each session, the group was moved from the home pen to the waiting area. For each trial, an individual goat was gently pushed into the start box (1 × 1 × 1 m) by experimenter 1 (E1). In the interest of standardizing the experimental conditions, the goat remained in the start box for 10 s before a transparent acrylic guillotine door was lifted to allow entry into the experimental area (2.9 × 1.4 m).

After entering the experimental area, the subject was allotted 60 s to retrieve the reward, which was presented in the rear section of the experimental area (Fig. 1). A piece of uncooked pasta (penne) was used as a reward in all phases of the experiment (Nawroth, Borell & Langbein, 2014). After the goat retrieved the reward or after 60 s elapsed without reward retrieval, the left or right door at the rear end of the experimental area was opened remotely, and the goat was led into one of the return alleys by experimenter 2 (E2) to independently re-join the group in the waiting area via the left or right return alley. Within one session, each subject underwent two to four trials. The order of testing within consecutive trials was randomized, and the side of the return alley was counterbalanced for individual goats in consecutive trials. All trials during training and testing were videotaped for subsequent coding of behaviour (Panasonic WVCP500, Tamron 13VG2811ASIR-SQ lens, EverFocus EDRHD-4H4 HD-CCTV Hybrid DVR).

Cylinder task

Apparatus

For the cylinder task, an opaque (shaping and training) or otherwise identical transparent (test) cylinder (20 cm in length, 17.2 cm in diameter, and 5 mm in thickness) was used. Each cylinder was open on both sides and mounted horizontally on a wooden platform (42 cm in height) (Fig. 2). The wooden platform was fixed to the ground to maintain the cylinder in place throughout the trials. During shaping only, two bowls (8 cm diameter) were attached at the openings of the opaque cylinder to encourage the goats to explore the cylinder (Fig. 2A).

Figure 2 Cylinders used in the different experimental phases.

Cylinders used in the different experimental phases. (A) Shaping. (B) Training. (C) Test.

Procedure

Shaping

Shaping was conducted to habituate the subjects to being alone in the experimental area and to induce them to approach the cylinder to obtain a reward. In a total of 16 trials, E2 baited both external bowls (Fig. 2A) with one piece of pasta each and left the experimental area before the goat entered the start box. The goat was released into the experimental area after 10 s. If the goat did not retrieve at least one reward within 60 s, E2 entered the experimental area and offered the pasta by hand. Two goats were excluded from the experiment at this stage. One refused to feed on the pasta, and the other showed extreme signs of arousal upon being left alone in the experimental area. All of the remaining goats directly approached the cylinder and ate the pasta from both bowls by the end of shaping.

Training

During training, the animals should learn to retrieve the reward from inside the opaque cylinder (Fig. 2B). In the first five training trials, the pasta was placed at the left or right edge of the cylinder, whereas in the subsequent trials, the reward was placed in the middle of the cylinder. For baiting, E2 was standing behind the cylinder when the goat entered the start box. When the goat looked through the transparent door in the direction of the cylinder, E2 made a short, distinct pointing gesture with her left or right arm and placed the pasta in the cylinder (see Video S1). The side from which E2 baited the cylinder was counterbalanced across trials and pseudo-randomized so that the cylinder was not baited twice in succession from the same side. Then, E2 left the experimental area. Each goat was released into the experimental area after 10 s. The criterion for admission to the test phase was successful retrieval of the reward within 60 s in six consecutive trials. All but one animal fulfilled the criterion within 12 trials. One goat required 16 trials to reach the criterion.

Test

The testing procedure was identical to the final training procedure with the exception that the transparent cylinder was substituted for the opaque cylinder (Fig. 2C). The test consisted of 10 trials per animal (see Video S2).

Data scoring and analysis

Behavioural coding was performed with the video footage using The Observer 12.0 (Noldus Information Technology, Wageningen, Netherlands). As two goats were excluded from the experiment during shaping (see above), the data from 20 goats were analysed. For the test trials, the latency to retrieve the reward (‘latency’) and the accuracy of the approach to the cylinder (‘accuracy’) were recorded. ‘Latency’ was defined as the length of time from the first step of the goat into the experimental area to its touch of the reward. Trials were rated as accurate (0) when the goat detoured to one side of the cylinder without touching the exterior of the cylinder. By contrast, trials were rated as inaccurate (1) when the goat tried to approach the reward directly by touching the front or back of the cylinder prior to retrieving the reward. In a study of human infants, Noland & Rodrigues (2012) argued that only those touches of a transparent surface that correspond to the reward’s position behind it express inhibition errors. I rated a trial as incorrect only when the goat touched the cylinder near the reward and not when it briefly explored the edge of the cylinder (see Video S2). For each trial, ‘side’ (left or right) was recorded, defined as the side by which the goat detoured the cylinder, and ‘TD-cyl’ was calculated, defined as the total duration over which the goat touched the exterior of the cylinder. To assess inter-observer reliability, a second observer who was not involved in the study recorded all of the behavioural data of the animals for 25% of the test trials. Cohen’s kappa indicated excellent agreement between coders across all recorded behavioural data (k = 0.98, p < 0.001; The Observer 12.0).

Statistical analyses were performed using the SAS System for Windows (SAS 9.4, TS Level 1M3, 2012). Generalized linear mixed models (PROC GLIMMIX) fitted for binary data were constructed to investigate the impact of test trial on ‘accuracy’ and the effect of the short distinct pointing gesture made by the experimenter during baiting on ‘side’. In the model, the distribution of the appropriate response variable (binary distribution) and the link function (logit) was specified, and a general Satterthwaite approximation was used for the degrees of freedom of the denominator. Individual animal was treated as the subject for the repeated statement and was considered in the factor trial. The effects of test trial on ‘latency’ and ‘TD-cyl’ were investigated by conducting repeated-measures ANOVA using PROC MIXED. Individual animal was treated as subject for the repeated statement and was considered in the factor trial. Least-squares means (LSM) and their standard errors (SE) were calculated for the variables of interest in all models. Where significant main effects were found (p < 0.05), adjustments for multiple testing were applied (Tukey–Kramer correction) in subsequent multiple comparison procedures. The binomial test was employed to detect individual side biases in detouring the cylinder. In addition, a continuous laterality index was calculated for each subject to test its correlation with ‘accuracy’ in test trials (Hopkins, 1999).

Results

The average accuracy during the test was 62.5% (±10.86%). There was no impact of trial number on accuracy (F9,171 = 0.85, n.s.). The accuracy of the individuals was variable across trials (Fig. 3). While nine goats detoured the cylinder correctly in seven or more trials, 11 animals touched the exterior of the cylinder in approximately every other trial.

Mean contact time with the cylinder (TD-cyl) in the test was 1.28 s (±0.64 s). There was no impact of trial number on TD-cyl (F9,171 = 0.93, n.s.). In most of the inaccurate trials, goats started detouring towards one of the open ends of the cylinder but then changed direction and briefly touched the near side of the cylinder in the region of the reward before finally detouring to the side and retrieving the reward (see Video S2).

Figure 3 Individual test trial accuracy.

Accurate (contact = 0) and inaccurate trials (contact = 1) of individual goats in the testing phase. Next to the number of the animal, each graph is marked with a different letter (from A to T).

In contrast to its impact on TD-cyl, trial number had a significant impact on latency to retrieve the reward in the test (F9,134 = 7.85, p < 0.001) (Fig. 4). Latency was 34 s (±3.62 s) in trial one and decreased to below 10 s from trial four onwards. Pairwise comparisons revealed latency in the test to be longer in trial one than in all subsequent trials (all p < 0.05). Furthermore, latency was longer in the first test trial than in the last training trial (p < 0.01).

Figure 4 Latency to retrieve the reward.

Mean latency (s, LSM ± SE) to retrieve the reward in the test and in the final training trial. Significant differences between trials are indicated by asterisks (*p < 0.05, **p < 0.01).

A short, distinct pointing gesture made by the experimenter during baiting had no effect on the side (left/right) by which the goats detoured the cylinder to retrieve the reward (F1,151 = 0.22, n.s). Ten out of 20 subjects (50%) showed an individual side bias in the test (p < 0.05). Among these subjects, six had a preference for detouring the cylinder to the left, and four preferentially detoured to the right. There was no correlation between the index of laterality and the level of accuracy in the test (Spearman rank correlation (rS) =  − 0.10, p = 0.66, n = 20).

Discussion

The goats in this study were able to retrieve a reward by showing motor self-regulation at a level comparable to or better than that of many other mammal and bird species tested to date (MacLean et al., 2014, Support. Inform., Table_S05; Vernouillet et al., 2016). Only great apes, some social corvids and various canine species have shown fundamentally better performance than the studied goats in the cylinder task (MacLean et al., 2014; Fagnani et al., 2016; Kabadayi et al., 2016; Chappell, 2017). However, when plotting the relationship between absolute brain size and performance in the cylinder task for 25 previously studied mammal species and the goat, the value for goat lies slightly below the regression line (Fig. 5). Unfortunately, to date, almost exclusively primate and canine species have been examined using this task. It would be of particular interest to study other taxa; for example, it would be of interest to study several herbivorous species to determine the effects of different feeding strategies or different social systems on their levels of behavioural control.

Figure 5 Relationship between brain size and motor self-regulation.

Relationship between absolute brain size and performance in the cylinder task for 26 mammal species from four orders. The trend line is based on a regression across all species. The two domestic animals species are marked in red. (1, Mongolian gerbil; 2, marmoset; 3, fox squirrel; 4, golden-headed lion tamarin; 5, mongoose lemur; 6, black lemur; 7, ring-tailed lemur; 8, squirrel monkey; 9, brown lemur; 10, Coquerel’s sifaka; 11, red-bellied lemur; 12, ruffed lemur; 13, aye aye; 14, capuchin monkey; 15, coyote; 16, domestic dog; 17, rhesus macaque; 18, golden snub-nosed monkey; 19, grey wolf; 20, domesticated goat; 21, hamadryus baboon; 22, olive baboon; 23, bonobo; 24, chimpanzee; 25, orangutan; 26, gorilla [MacLean et al., 2014; Ballarin et al., 2016]).

It has been demonstrated that monkey species living in fission–fusion societies, which are based on individual recognition, social cooperation and pair bonding, tend to show higher levels of motor self-regulation than do species living in more stable groups (Amici, Aureli & Call, 2008). Feral and domestic goats have been shown to live in fission–fusion societies (Stanley & Dunbar, 2013; Ævarsdóttir, 2014), to generate individual contact calls in both kids and their mothers for individual recognition (Briefer & McElligott, 2011; Briefer, Padilladela Torre & McElligott, 2012), and to easily discriminate between members of their own group and those of a different group (Keil et al., 2012). According to its feeding ecology, the goat is classified as an ‘intermediate grazer’, meaning it exhibits a marked degree of foraging selectivity (Stuth, 1991). These aspects of the goat’s social life, cognitive abilities and feeding ecology may explain its apparently good performance in motor-self regulation. However, I found large inter-individual variation among animals in the level of motor self-regulation. Only approximately half of the subjects exhibited high levels of behavioural inhibition. Nine out of the 20 goats were able to consistently suppress the prepotent response of directly approaching the visible reward in seven or more trials, whereas the remainder exhibited poor motor control. These latter goats approached the reward directly in approximately every other trial. There was no improvement of motor self-regulation at the group level over the testing period.

Previous studies using the cylinder task have reported inconsistent results regarding the learning of motor self-regulation. While MacLean et al. (2014), who compared motor self-regulation in 36 species, did not report data on improvement over test trials, other authors have analysed performance across trials to evaluate potential learning effects. No such effects on motor self-regulation were found in some studies of dogs and wolves (Bray, MacLean & Hare, 2014; Marshall-Pescini, Viranyi & Range, 2015). By contrast, trial number was found to have an effect on motor-self regulation in dogs in one study and in some bird species, indicating that learning across trials occurred (Fagnani et al., 2016; Kabadayi et al., 2016; Vernouillet et al., 2016). Most recent studies have compared the level of accuracy between only the first and last blocks of five trials. It has been suggested that an improvement in motor self-regulation across trials might indicate insufficient training with the opaque cylinder (Santos, Ericson & Hauser, 1999; Smith et al., 1999) or a lack of experience with transparent surfaces (Yates & Bremner, 1988; Vernouillet et al., 2016); however, no such relation was found by Fagnani et al. (2016) in dogs. Both explanations can be excluded for the goats in this study. They performed well with the opaque cylinder after only a few training trials, and they had experience with transparent surfaces, including those in a prior maze test (see ‘Animals, materials & methods’) and the transparent doors in this study (see Fig. 1). Furthermore, nearly all of the goats accurately detoured the cylinder in 50% of all trials in the test, indicating that any inaccurate trials did not result from a lack of knowledge about how to solve the task or the concept of transparency. It seems that for the majority of the goats, the visual salience of the reward inside the transparent cylinder was sufficiently strong to overpower the previously learned accurate motor pattern used to retrieve the reward. Some researchers have argued that the visibility of the reward behind the transparent barrier acts as a ”magnet for perception”, making it very difficult for animals to avoid direct approach and activate the learned behaviour pattern of detouring (Vallortigara & Regolin, 2002). Similar effects have been shown in various object retrieval tasks in human infants at the age of 7 months (Diamond, 1981; Diamond, 1990). Therefore, errors during test trials appear to result from failure of reliable motor self-regulation. However, I cannot exclude the possibility that the accuracy of the goats in the test would have improved after more trials. In a study with song sparrows, some individuals reached the learning criterion of six correct trials in succession in the cylinder task only after 50 trials (Boogert et al., 2011). However, this outcome is unrelated to behaviour control; rather, it shows that some sparrows learned a new behaviour pattern that allowed them to overcome the impulse to directly approach the reward.

Unfortunately, only one other study on motor self-regulation has been performed in which latency to retrieve the reward was investigated; the subjects were various parrot species (Kabadayi et al., 2017). According to the authors of that study, a reduction in latency is generally seen as a sign of learning the task. In the current study, the trial number in the test had a significant effect on the latency to retrieve the reward. However, this effect was mainly caused by the latency in the first test trial, which was significantly longer than that of any of the subsequent trials. Additionally, latency was significantly longer in the first test trial than in the final training trial. The goats reacted with great caution to the introduction of the transparent cylinder in the test. Novel objects are known to be fear-inducing stimuli in ungulates (Désiré et al., 2004). Therefore, I believe that the goats’ initial fear of the transparent cylinder rather than a learning effect was responsible for reducing latency over the first trials. Owing to this fear of new objects, one would expect animals performing the cylinder task to carefully approach and investigate the exterior of the transparent cylinder before attempting to retrieve the reward inside, especially in the first few trials. Therefore, initial fear could indirectly affect the number of inaccurate trials during the test. This possibility is important to consider in future detour-reaching studies. However, the goats in this study rapidly overcame their initial fear of approach as indicated by the rapid decrease in latency. Additionally, as discussed above, the number of inaccurate trials was not affected by the trial number.

Detour tasks are widely accepted as the most suitable tasks for comparing motor self-regulation across species. However, recently, there has been increasing criticism of the general validity of this task resulting from the large variation among species and the large inconsistency in performance among different detour tasks within species (Kabadayi et al., 2017; Kabadayi, Bobrowicz & Osvath, 2018; Van Horik et al., 2018). Individual performance in detour tasks, especially the cylinder task, may be confounded by various accompanying factors, such as the level of neophobia of novel objects (Regolin & Vallortigara, 1994), level of experience with transparent surfaces as barriers and learning effects. Although I have discussed the influences of some of these factors in detail, it would be valuable to investigate the specific aspects of various detour tasks within the framework of task batteries, as has been done in some recent studies (Amici, Aureli & Call, 2008; Brucks et al., 2017).

There is ongoing debate as to whether and how animals perceive and process human-given social cues, such as pointing, to indicate the location of a reward or to direct an animal’s movement (Tauzin et al., 2015). Among domestic animals, dogs (Pongracz et al., 2013) and various farm animals, such as goats (Kaminski et al., 2005; Nawroth, Borell & Langbein, 2015b) and horses (Proops, Walton & McComb, 2010; Lovrovich, Sighieri & Baragli, 2015), have been shown to make use of human-given cues. However, one should not overestimate the cognitive abilities necessary to respond appropriately to human pointing, as most experiments involving human pointing did not differentiate between local or stimulus enhancement and actual referential comprehension of the task. For monkeys and horses, pointing worked well when the pointing finger, hand or arm remained close to the target until the animal made a choice (Maros, Gacsi & Miklosi, 2008; Schmitt, Schloegl & Fischer, 2014). A more challenging form of human social cues is momentary pointing from a distance (Gácsi et al., 2009). In the present study, the experimenter made only a short, distinct pointing gesture to indicate the side from which the cylinder was baited before the subject was allowed to make a choice. Dogs and elephants have been shown to be capable of using social cues as referential signals during momentary pointing (Pongracz et al., 2013; Smet & Byrne, 2014). With the goats in the present study, there was no evidence that the short, distinct pointing gesture during baiting had an impact on the side by which the goats detoured the cylinder. Therefore, I do not believe that the pointing gestures had any influence on the level of motor self-regulation in this study.

Different types of detour tests, aside from those studying motor self-regulation, have been employed to investigate lateralization in several species under natural and experimental conditions (Vallortigara, Regolin & Pagni, 1999; Baragli et al., 2011; Leliveld, Langbein & Puppe, 2013; Siniscalchi, Pergola & Quaranta, 2013). Laterality refers to the phenomenon in which external attractions are perceived and processed differently by the two cerebral hemispheres depending on their novelty and emotional value and in which the execution of motor behaviour is preferentially performed by one side of the body. Owing to lateral eye position and the decussation of optic nerve fibres at the optic chiasm in ungulates (approximately 80–90% in large domestic ungulates Shamir & Ofri, 2008), visual cues perceived by the left eye are largely, though not exclusively, processed by the right hemisphere and vice versa. In a recent study in goats (Nawroth, Baciadonna & McElligott, 2016b), the authors did not find agreement over repeated trials regarding the side to which goats detoured a transparent barrier. By contrast, half of the animals in this study showed individual side biases in detouring the transparent cylinder in the test. An approximately equal number of goats preferred either side. This result indicates that at least some of the goats showed lateralization of the detour behaviour at the individual level, but such lateralization was not evident at the population level. Similar results have been found regarding lateralization of detour behaviour in sheep (Versace et al., 2007). However, there was no impact of laterality on the level of motor self-control in goats in this study.

Conclusions

The results demonstrate that goats display motor self-regulation at a level comparable to or better than the levels observed in many other bird and mammal species tested to date. However, the goats did not show any improvement in the level of motor self-regulation across trials and showed large intra- and inter-individual variability across test trials. The results indicate the importance of considering individual reaction patterns when analysing detour behaviour.

Supplemental Information

Supplemental Information 1 Raw data

Click here for additional data file.

Supplemental Information 2 Cylinder task - training trial

The full procedure of a training trial.

Click here for additional data file.

Supplemental Information 3 Cylinder task test trial

The full procedure of a test trial. The first sequence shows an accurate trial and the second sequence shows an inaccurate trial.

Click here for additional data file.

I would like to thank Katrin Siebert for data coding, Armin Tuchscherer for statistical advice, and Dieter Sehland and Heinz Deike for excellent technical assistance. I would also like to thank Christian Nawroth for in-depth discussions and valuable comments on an earlier version of the manuscript. Finally, we thank Lucia Regolin and two other anonymous reviewers for their helpful comments on an earlier version of this manuscript.

Additional Information and Declarations

Competing Interests

Author Contributions

Animal Ethics

Data Availability

The authors declare there are no competing interests.

Jan Langbein conceived and designed the experiments, performed the experiments, analyzed the data, contributed reagents/materials/analysis tools, prepared figures and/or tables, authored or reviewed drafts of the paper, approved the final draft.

The following information was supplied relating to ethical approvals (i.e., approving body and any reference numbers):

All procedures involving animal handling and treatment were approved by the Committee for Animal Use and Care of the Ministry of Agriculture, Environment and Consumer Protection of the federal state of Mecklenburg-Vorpommern, Germany (Ref. No. 7221.3-2-012/15 and 7221.3-2-011/16).

The following information was supplied regarding data availability:

The raw data have been provided in Supplemental Files.

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
