# Peer review of "Motor self-regulation in goats (Capra aegagrus hircus) in a detour-reaching task"

_PeerJ, doi:10.7717/peerj.5139_

## Round 0.1 · original submission · Minor Revisions

The reviewers liked your paper but required some amendments and clarifications. Please address them and re-submit the paper.

Reviewer 1 ·

Basic reporting

I recently reviewed the same study for another journal. The author addressed most of my suggestions in the current manuscript, however apparently he still missed an important study that already looked at the reduction of latency in retrieving the reward across trials on the cylinder task (Kabadayi et al. 2017, “Are parrots poor at motor self-regulation or is the cylinder task poor at measuring it?”). Thus, the sentence beginning in line 319 is incorrect, and this study is not the first one that investigated the trial effect on latency to obtain the reward on the cylinder task.

Experimental design

No comment.

Validity of the findings

The author cites a large-scale study that measured motor self-regulation on 36 species, using the cylinder task (MacLean et al. 2014). Since the author can only interpret the 65% success rate of the goats in relation to this large-scale study, it would be helpful to compare the performances of the goats with those 36 species (or the meaningful subset of those species) in a graph or a table. In the absence of such comparisons, it becomes confusing for the reader to assess how well goats did in relation to other species – thus the aim of the study.

Besides, MacLean et al. (2014) found absolute brain size was a significant predictor of the cylinder task performance among 36 species. The author then should then discuss the apparently high performance of the goats in relation to the absolute brain size of these species. For example, Kabadayi et al. (2016) renewed the top 10 list of MacLean et al. after the inclusion of three bird species they tested, and they included a figure showing the relationship between the absolute brain size and the cylinder task performance within 10 bird species.

Which was the species in MacLean et al. that was most closely related to goats, and how well did they do? 65% success rate of the goats in this study only makes sense after such close comparisons and analyses. Without such closer investigations, and with using vague terms such as 'low', 'intermediate' and 'high' in describing the cylinder task scores, the reader finds it hard to get the main point of the study (the comparative investigation of motor self-regulation).

Additional comments

Lines 81-83, the author puts a question whether inhibitory control is a unitary phenomenon or different tasks measure different aspects of it, however the study only administers one task, leaving these discussions rather redundant and unanswered. Besides, there is also evidence that detour reaching tasks do not purely measure motor self-regulation as implied in ln 90, because there are various potential factors affecting the performance, such as the previous experience with transparency, rule learning and functional generalization from the opaque barrier to the transparent barrier (Kabadayi et al. 2018). Given the motivational and emotional factors, age, as well as the level of previous experience with transparency affects the detour task performance, why did the author not suggest task batteries in the future as a more beneficial way to assess motor self-regulation in goats, or acknowledge the limitation of using a single task in the current study?

I welcome the investigation of the trial/learning effect on goat performances in this study, however the author misses an important study that already looked at the reduction of latency in retrieving the reward across trials on the cylinder task (Kabadayi et al. 2017, “Are parrots poor at motor self-regulation or is the cylinder task poor at measuring it?”). Thus, the sentence beginning in line 319 is incorrect. Although the author did not find a learning effect on “accuracy” within 10 trials, administering more trials could eventually lead to a ceiling effect. For example, Boogert et al. (2011) found that although not detected within 10 trials, trial effect was eventually found in song sparrows after around 50 trials where their performances peaked. The authors should discuss this possibility for goats too.

In fact, a study with parrots (Kabadayi et al. 2017) questioned the efficiency of the cylinder task as a measure of motor self-regulation, especially when different species are compared only with the terminal scores obtained within 10 trials, with the currently very strict coding criterion (every contact with the transparent cylinder counts as incorrect, regardless whether the touch was aimed towards the reward behind the barrier). The author might consider analyzing the incorrect trials for whether the touch to the barrier was directed towards the reward. Kabadayi et al. (2017) precisely looked at this with parrots, and found that two parrot species’ failures were not ‘food directed’, i.e. they touched the barrier ‘away’ from the reward. This implies they were merely exploring the transparent cylinder rather than trying to reach for the visible reward, pointing to yet another weakness of the cylinder task and its strict coding criterion (all touches to the barrier counts as a motor self-regulation failure). Conducting similar analyses on the failure patterns with goats would strengthen the study.

Lines 60-64: Why does this discussion come in the second paragraph and not in the first?

Ln 83: An important empirical study that is relevant to this is Brucks et al. 2017 in Frontiers in Psychology: 'Measures of Dogs’ Inhibitory Control Abilities Do Not Correlate across Tasks'

Ln 97: Remove 'must'

Ln 106-108: What is meant by 'high' and 'low'? This is critical for the whole interpretation of the goat performance. Is 65% high or low? What is 'intermediate' performance? Is it 50%? This is the reason why I suggest presenting the results of the goats together with the other species on a table or a graph, so that the reader gains an insight on the level of motor self-regulation in goats in relation to other species.

Ln 297: It was Kabadayi et al. (2016) that analysed the trial effect for bird species tested in MacLean et al. (2014). Three corvid species tested in Kabadayi et al. (2016) did not show any trial effect.

·

Basic reporting

I enjoyed reading this article, it is very well written. The experiments are accurately described and I did not notice any missing information for fully understanding the study.

Experimental design

The experimental design is flawless, as it is based on a consolidated (and rather straightforward) task run before in several other species.

Validity of the findings

Data and statistical analysis are convincing. Findings are interesting and nicely discussed, conclusions are sound.

Reviewer 3 ·

Basic reporting

1) Despite I’m not English mother language I would suggest author to carefully check the manuscript with the help of a native English speaker. In several parts the sentences and the context regarding a specific paragraph appear disentangled with each other. This make difficult to follow the manuscript’s reasoning (i.e. see lines 115, 117, 119-123, 268, 313-320, 333)
2) The introduction section is not well defined. The Author uses the proper citations but the rationale is weak and does not provide a clear frame for the hypothesis (which is lacking). It is undeniable that researching and comparing on new species can help to build a more structured overview on the argument, but the author should formulate his statement on a strong concrete premise. For example, which are the assumptions on social and cognitive abilities in goats, on which, by the way, the expected results seem to be based?
3) The Introduction should be shortened a bit. Qualitative descriptions of unfitting tests are unnecessary to the aim of the study (lines 78-89) and could be avoided. It could be more useful if the author focuses on why detour (cylinder task) properly fits with the aim of the purpose of the study. For example the detour/cylinder task may be introduced at the beginning, followed by a clear explanation as to why this task seems to be the best one for studying motor self-regulation in goats (by moving earlier lines 91-95).
4) In my opinion, the greater issue consists in the fact that, in the present form, the manuscript lacks in a clear hypothesis. The assumptions on which this work seem to lie on are: “broad the range of specie” (lines 104-106); “in addition it was investigated whether short, distinct pointing gestures made by the experimenter during baiting would influence the side of the cylinder to which the goats detoured” (lines 113-115) and “it was analyzed whether the goats would show a side bias in detouring the cylinder” (lines 116-117). Despite lines 117-119, what the author expects and why is not clear to me. Saying a few more words could help to define the rationale behind this work.
5) Line 21, 48, 66 and others: giving the meaning of “prepotent” I’m not sure this word is properly used here.
6) Figure 3 is complex and not easy to read. Maybe such data can be shown in Table.

Experimental design

1) The experimental design mixes two different paradigms (detour and human pointing). This is not an error “per se”, but the author should be aware that the submitted results may be the natural consequence of the animal’s evaluation of the entire sets of stimuli. More broadly, could the presence of a human have affected in some way the behavior and the latency to reward? May the human pointing give us some adjunctive information about animals taking into account to solve (or not solve) the task? I suggest the author to clearly describe this mixing procedure in the Animals, material & methods section. Furthermore I would suggest to not considering his results as separate outcomes, adding some sentences in order to discuss them together (see points below).
2) On line 199 author states that the side of human pointing was counterbalanced across trials. Which Criterion was used in this case?
3) On lines 188-189 author states that if the goat did not retrieve at least one reward within 60 s, the experimenter offered the reward by hand. This means that the behavior performed by the goat at that time has been positively reinforced (i.e. if goat was standing at the end of 60 sec, this behavior was reinforced). Does not the Author believe that this procedure could have affected the results in some way?

Validity of the findings

Regarding the discussion, I would suggest the author to spend some few more words to discuss together his results by mixing the two different paradigms studied (detour + pointing). Regarding that, I would recommend to expend references (i.e. Lovrovich et al 2015 and Baragli et al 2017 and similar). Therefore he should include additional citations to discuss together the results obtained. On line 278-279 author declare that, at the individual level, goats showed ambiguous results. Could such results have been affected by factors linked to individuals? For example, by personality or the way through animals perceive and use human pointing. See lines 282-283: could have the shy/bold traits affected results at individual level?

Additional comments

Dear Editor
I have carefully read the paper titled “Motor self-regulation in goats (Capra aegagrus hircus) on a detour-reaching task” in which author designed a specific test mixing detour and pointing paradigms, with the aim of studying inhibition of motor control in goats. Despite the topic is worthy of interest and the experiment looks generally well designed, I have found some major issues that should be solved before considering this paper for publication.

Annotated reviews are not available for download in order to protect the identity of reviewers who chose to remain anonymous.
External reviews were received for this submission. These reviews were used by the Editor when they made their decision, and can be downloaded below.

---

## Round 0.2 · accepted · Accept

The reviewer is satisfied with your revision, and after reading myself the paper I concur with this evaluation. I am happy to accept your paper.

# Reviewer 3 ·

Basic reporting

No further comments.

Experimental design

The author replies to all of my concerns and I’m satisfied about his reply.

Validity of the findings

The paper expands the knowledge on topic.

Additional comments

I’ve no further comments.